# External validation of the Meggitt-Wagner, Texas University, SINBAD, and Saint Elian classifications for predicting major amputation in patients with diabetes at a public hospital in Peru

Luis Alberto Gallardo-Alburqueque[1], Yudith Quispe-Landeo[2], Ann Chanamé-Marín[3], Leonardo J. Uribe-Cavero[4], Marlon Yovera-Aldana[3]*

1 Universidad Científica del Sur, Lima, Perú, 2 Universidad San Martin de Porres, Lima, Perú, 3 Grupo de Investigación NEMECS: Neurociencias, Metabolismo, Efectividad Clínica y Sanitaria, Universidad Científica del Sur, Lima, Perú, 4 Sociedad Científica de Estudiantes de Medicina de Ica (SOCEMI), Facultad de Medicina Humana, Universidad Nacional San Luis Gonzaga, Ica, Peru

* myovera@cientifica.edu.pe

## Abstract

### Objective

To externally validate and compare the prognostic performance of four diabetic foot ulcer classifications—Meggitt-Wagner, University of Texas, SINBAD, and Saint Elian— for predicting major amputation in patients with diabetes treated at a public hospital in Peru.

### Materials and methods

We conducted a retrospective cohort study at María Auxiliadora Hospital, one of the few referral centers in Lima with a specialized Diabetic Foot Unit. The study period was January 2015 to December 2019. Eligible patients had a lower-limb ulcer, complete clinical data recorded within 48 hours of the index healthcare encounter (hospital admission or outpatient evaluation), and at least six months of follow-up. Patients with venous ulcers or pressure ulcers related to immobilization were excluded. The primary outcome was major amputation, defined as any procedure above the ankle. Prognostic performance of the four systems was assessed using the area under the receiver operating characteristic curve (AUROC), sensitivity, specificity, predictive values, and the Youden index

### Results

A total of 342 patients were included. All had type 2 diabetes mellitus, median age was 62 years (IQR 54–69), and 229 (67.0%) were male. The six-month cumulative incidence of major amputation was 11%. The AUROC values were: Saint Elian, 0.90; University of Texas (3D stage), 0.81; Meggitt-Wagner, 0.81; and SINBAD, 0.74. The

**Data availability statement:** All relevant data are within the paper and its Supporting information files.

**Funding:** This research received no specific funding. If accepted for publication, the article processing charge (APC) will be funded by Universidad Científica del Sur, Peru. The funder will have no role in study design, data collection and analysis, decision to publish, or preparation of the manuscript.

**Competing interests:** The authors have declared that no competing interests exist.

**Abbreviations:** SE, Saint Elian; MW, Meggitt-Wagner; TU, Texas University score; AUROC, Area under the receiver operating characteristic curve; MINSA, Peruvian Ministry of Health; SIS, Seguro Integral de Salud; IWGDF, International Working Group on the Diabetic Foot

most discriminative thresholds were SE ≥ 18, TU stage 3D, MW ≥ 3, and SINBAD ≥5. Patients with SE > 20 had a 32-fold higher risk of amputation compared with those with SE < 16.

## Conclusion

The Saint Elian and University of Texas systems showed the best prognostic accuracy, while Meggitt-Wagner and SINBAD performed moderately. These findings provide clinicians with clear cut-off points to identify high-risk patients and support early interventions in referral settings. Their effective use depends on multidisciplinary teams trained in diabetic foot management, and selecting the classification system most suited to hospital resources and patient profiles may help reduce amputation rates.

## Introduction

Diabetic foot is one of the most frequent and severe complications of diabetes mellitus. A lower limb amputation due to diabetic foot occurs every 20 seconds worldwide, making it the leading cause of non-traumatic limb loss [1]. Patients with diabetic foot ulcers have a 2.5-fold increased risk of death within five years compared to those with diabetes but without ulcers [2]. The direct costs associated with diabetic foot account for approximately one-third of total diabetes-related expenditures and are comparable to the investments made in oncological diseases [3] Given its high burden, the study of its prevalence and prognosis is essential to improve clinical outcomes and optimize healthcare resource allocation [4].

In clinical practice, prediction rules based on signs, symptoms, medical history, and laboratory findings are widely used to assess disease prognosis, guide therapeutic decisions, and enhance communication between healthcare providers and patients [5]. In the case of diabetic foot, such tools are primarily applied to predict the likelihood of major amputation or the success of limb-salvage interventions [6]. Currently, more than 25 classification systems have been developed, each incorporating a different number of criteria [7]. Their performance varies depending on the available technology, the characteristics of the reference population, and the presence of multidisciplinary care teams [8].

In the Peruvian context, diabetic foot affects approximately 5.9% of patients attending healthcare facilities [9]. It is also accounts one out of every five hospital admissions among patients with diabetes in Peruvian hospitals [10]. Limitations in healthcare personnel training, infrastructure, and funding hinder the establishment of multidisciplinary teams dedicated to diabetic foot care. [11] Furthermore, only 11.5% to 15% of these patients achieve adequate metabolic control —defined as optimal levels of blood glucose, blood pressure, and lipid profile —which further contributes to suboptimal clinical outcomes [12,13].

Most clinical scoring systems for diabetic foot were developed and validated in high-income countries, and their predictive performance may not translate directly to

Latin American settings. Differences in patient profiles, healthcare pathways, and resource availability can influence prognostic accuracy, underscoring the need for external validation. [14] To address this gap, we compared the discriminative capacity of four widely used classifications—Meggitt-Wagner (MW), University of Texas score (TU), SINBAD, and Saint Elian (SE)—to predict major amputation at six months among patients managed at the Diabetic Foot Unit of a national referral hospital in Peru.

## Methods

### Study design

We performed an external validation study using a retrospective cohort design, analyzing prospectively recorded clinical data from the Diabetic Foot Unit at María Auxiliadora Hospital (Lima, Peru) between January 2015 and December 2019.

### Study setting and clinical management

María Auxiliadora Hospital is a public referral center located in southern Metropolitan Lima, serving 13 districts (~2.5 million inhabitants). It primarily attends an insured population covered by Peru's state-subsidized program for low-income and very low-income groups (Seguro Integral de Salud, SIS). Patients with diabetes and lower-limb ulcers, infection, or ischemia are assessed by the Diabetic Foot Unit either from outpatient services or the emergency department. The Unit applies a standardized diagnostic-therapeutic pathway: initial debridement with tissue sampling and targeted antibiotics, an intensified wound-care schedule during the acute phase (daily, Monday to Saturday), step-down to three sessions per week upon stabilization, vascular assessment with arterial Doppler waveforms, and inpatient or ambulatory follow-up until complete epithelialization. After epithelialization, patients continue monthly follow-up on an ongoing basis to assess for recurrence.

### Population, sample and sampling

Eligible patients were identified from the prospectively maintained Diabetic Foot database of the Endocrinology Service, which records all patients managed by the Diabetic Foot Unit. We included patients with a lower-limb ulcer and complete baseline clinical information documented within 48 hours of the index healthcare event (hospital admission or outpatient evaluation), with at least six months of follow-up. Diabetic foot was defined as a lower-extremity ulcer in a patient with documented diabetes mellitus (medical history and laboratory records). We excluded venous ulcers (typical location and signs of chronic venous insufficiency, without neuropathic/ischemic features) and pressure ulcers related to immobilization (heel location with bedridden status). Patients with neuropathic, ischemic, or mixed diabetic ulcers were included. A non-probabilistic, convenience sampling approach was used, based on consecutive cases in the database during the study period.

### Variables

**Diabetic foot classifications.** We evaluated four widely used systems—Meggitt-Wagner (MW), University of Texas (TU), SINBAD, and Saint Elian (SE). MW and TU were included due to their historical and global use as reference frameworks; SINBAD is recommended by the IWGDF for international comparability; and SE, developed in Latin America, integrates multiple clinical domains relevant to resource-limited environments. Each system was applied as originally defined

**Meggitt-Wagner Classification (MW):** This system is primarily based on ulcer depth, with partial consideration of ischemia. It includes six grades: grade 0, intact skin; grade 1, superficial ulcer; grade 2, ulcer reaching tendon or joint capsule; grade 3, deep ulcer with osteomyelitis or abscess; grade 4, localized gangrene (e.g., toes or forefoot); and grade 5, extensive gangrene involving the entire foot [15].

**Texas University Classification (TU):** This system incorporates three dimensions: ulcer depth, infection, and ischemia. Depth is graded from 0 to 3: (0) pre- or post-ulcerative lesion with intact skin, (1) superficial wound not involving tendon, capsule or bone, (2) wound penetrating to tendon or capsule, and (3) wound penetrating to bone or joint. The presence of infection and/or ischemia is categorized as follows: (A) neither present, (B) infection, (C) ischemia, (D) both infection and ischemia. These combinations yield 16 possible grades with distinct prognostic implications for limb salvage and amputation [16].

**SINBAD Classification:** Developed by Ince et al. in 2008 as a modification of the (S[AD])SAD system, it is an acronym that evaluates six parameters: *Site* (forefoot vs. midfoot/hindfoot), *Ischemia* (yes/no), *Neuropathy* (yes/no), *Bacterial infection* (yes/no), *Area* (<1 cm² or ≥1 cm²), and *Depth* (superficial or involving muscle/bone). This simplified score is intended for broad applicability in clinical and resource-limited settings [17].

**Saint Elian Classification (SE):** This system is composed of ten parameters organized into three domains [18]. A cumulative score is generated, classifying severity as mild (<10 points), moderate (11–20 points), or severe (21–30 points).

- *Anatomical factors*: location (phalanges, metatarsal, or tarsal), topography (dorsal/plantar, lateral/medial, or involving two or more areas), and number of affected regions (one, two, or the entire foot).

- *Aggravating factors*: ischemia (none, mild, moderate, or severe), infection (none, mild, moderate, or severe), edema (none, perilesional, limited to the affected leg, or bilateral), and neuropathy (none, diminished sensation, complete loss of protective sensation, or presence of Charcot arthropathy).

- *Contributing factors*: depth (superficial, below dermis, or full-thickness), ulcer area (<10 cm², 10–40 cm², or >40 cm²), and healing phase (epithelialization, granulation, or inflammatory phase).

**Outcome definition and ascertainment.** The primary outcome was major amputation, defined as any amputation above the ankle (infracondylar/supracondylar levels). Only amputations ipsilateral to the index ulcer were considered. Minor amputations (toe or transmetatarsal level) were classified within the "no major amputation" group. Outcomes were ascertained from hospital surgical records with active clinical follow-up; for missed visits, patients or relatives were contacted by telephone. Because this was a retrospective validation based on routinely collected data, predictors and outcomes were recorded in clinical practice prior to this analysis, and outcome assessors were not influenced by the classification scores. Given the absence of a national electronic medical record system during the study period, amputations performed at other institutions may have been missed; this is acknowledged as a study limitation.

**Other variables.** Covariates included sociodemographic, anthropometric/clinical, medical history, ulcer characteristics (type, size, depth, location), hospitalization status (outpatient vs. inpatient), length of hospital stay, ischemia (see below), and laboratory data (HbA1c, hemoglobin, LDL cholesterol, albumin). All patients in the cohort had type 2 diabetes mellitus; no cases of type 1 diabetes were recorded.

- **Sociodemographic variables:** Age in years (<60; ≥60), sex (male/female), and educational attainment (primary or less; secondary; higher education).

- **Anthropometric and clinical data:** Body mass index (BMI) in kg/m² (<25; 25–29; ≥30), diabetes duration in years (<10; ≥10), and type of diabetes treatment (diet only; oral antidiabetic drugs; insulin).

- **Medical history:** Presence or absence of hypertension, coronary artery disease, stroke, and history of prior major amputation.

- **Ulcer characteristics:** Type (new vs. recurrent), size (measured 48 hours after admission using the ellipse formula, categorized as <10 cm²; 10–39 cm²; ≥40 cm²), and depth (based on the IWGDF PEDIS classification). Ulcer location was classified as forefoot (yes/no).

- **Ischemia:** Ischemia was assessed by arterial Doppler waveform analysis (monophasic, biphasic, or absent flow in the anterior tibial, posterior tibial, or popliteal arteries). Although toe–brachial index (TBI) and ankle–brachial index (ABI) provide quantitative measures, they were often not feasible due to extensive lesions, edema, or pain, and their interpretation can be unreliable in patients with medial arterial calcification (Mönckeberg sclerosis). In such contexts, qualitative Doppler waveform assessment is endorsed in the literature as a valid alternative, since monophasic or absent waveforms correlate with significant ischemia and adverse limb prognosis. [19] While operator-dependent, this method remains practical and widely applicable in diabetic-foot care when pressure indices are limited. [20]

- **Laboratory data:** Glycated hemoglobin (HbA1c < 7%; ≥ 7%), hemoglobin (<8 g/dL; ≥ 8 g/dL), LDL cholesterol (<100 mg/dL; ≥ 100 mg/dL), and serum albumin (<2.5 g/dL; ≥ 2.5 g/dL).

## Procedures

Permission to access the diabetic foot database was obtained from the Endocrinology Service of María Auxiliadora Hospital. The dataset was provided in a fully anonymized format, with all personally identifiable information removed to ensure patient confidentiality

## Statistical analysis

The authors accessed and processed the dataset between December 1, 2022 and Augut 31, 2023. All analyses were conducted using R software, version 2025.05.0 + 496. A priori, statistical power was calculated to ensure that the available sample size was adequate for the planned comparisons, using a two-sided alpha level of 5%. The detailed results of the power analysis are presented in S1 Table. Missing data were assessed for all variables. No imputation was performed; analyses were conducted using complete-case data. All variables required to compute the four classification systems and the primary outcome had complete data (100% availability).

Descriptive statistics were reported using absolute and relative frequencies for categorical variables, stratified by the presence or absence of major amputation. Differences between groups were assessed using the chi-square test for nominal variables and the Wilcoxon rank-sum test for ordinal or non-normally distributed variables. Additionally, the median and interquartile range (IQR) of length of hospital stay were calculated.

The six-month cumulative incidence of major amputation was determined according to MW, TU, SINBAD, and SE. Relative risks (RR) with 95% CIs were calculated using the lowest theoretical-risk category as reference. To ensure statistical stability, categories with sparse or no events were collapsed with adjacent groups. For the Meggitt-Wagner classification, grades 1–3 were integrated because no amputations occurred in grades 1–2 and only three amputations were observed in grade 3. For the Texas University classification, most amputations were concentrated in stage 3D, while other stages had very few or zero events; therefore, categories were collapsed into "3D" versus "other stages." This merging approach was used to ensure sufficient sample sizes for statistical analyses and is consistent with prior validation studies evaluating diabetic foot classification systems [21].

Discrimination was assessed via AUROC with pairwise comparisons (DeLong test for correlated ROC curves) and additional sensitivity analyses were performed stratifying patients by their care setting (outpatient versus inpatient).. To facilitate interpretation of prognostic accuracy, we prespecified thresholds for the area under the ROC curve. An AUROC of 0.50 indicates chance-level discrimination; 0.51–0.59, very poor discrimination; 0.60–0.69, poor discrimination; 0.70–0.79, moderate discrimination; 0.80–0.89, good discrimination; and ≥0.90, excellent discrimination [22]. In addition to the primary outcome (major amputation), we conducted a complementary analysis in which the discriminatory performance of all classification systems was evaluated for the composite outcome "any amputation" (major or minor). This secondary analysis used the same ROC framework, with AUCs and 95% binomial exact confidence intervals estimated identically to the primary analysis.

Sensitivity, specificity, positive predictive value (PPV), and negative predictive value (NPV) were estimated for each classification and the category or score with the highest discriminative value was identified using the Youden index. A two-sided p-value < 0.05 was considered statistically significant.

### Reporting guidelines

This study was reported in accordance with the Strengthening the Reporting of Observational Studies in Epidemiology (STROBE) guidelines for cohort studies [23] and the Transparent Reporting of a multivariable prediction model for Individual Prognosis Or Diagnosis (TRIPOD) statement for external validation studies [24]. The completed checklists are available in Supplementary S1 Table (STROBE) and Supplementary S2 Table (TRIPOD).

### Ethical aspects

This study was approved by the Endocrinology Department of María Auxiliadora Hospital and the Institutional Ethics and Research Committee of the Universidad Científica del Sur (Approval Certificate No. 386-CIEI-CIENTÍFICA-2021). The ethical approval for the project was renewed on August 10, 2023, extending its validity for six months, until February 10, 2024. The analyzed database contained no personally identifiable information, as all data were encrypted. Only physicians from the hospital's endocrinology service had access to the identification codes linking patient data.

## Results

A total of 357 patients were recorded in the diabetic foot database. Fifteen were excluded because they had incomplete diabetic foot classification data (n = 12) or an unknown outcome (n = 3), resulting in 342 patients included in the analysis. Among these, 39 (11.4%) underwent a major amputation, while the remaining 303 did not; this latter group comprised 76 patients with a minor amputation (22.2%) and 227 with no amputation (66.4%). Overall, any amputation (major or minor) occurred in 115 patients (33.6%) (Fig 1).

Of 357 patients, 15 were excluded, leaving 342 for analysis, including 39 major amputations and 303 without major amputation (76 minor amputations and 227 without amputation).

### General characteristics

The average age of the population was 60 years, with two out of three being men. Hypertension was present in 38%, 3.9% had a prior stroke, and 7% had a previous major amputation. Diabetes duration was ≥ 10 years in 62%, 41% received insulin treatment before infection, 80% had HbA1c ≥ 7%, 12% had albumin < 2.5 g/dL, and 75% had hemoglobin < 8 g/dL. Regarding foot characteristics, 77% had an ulcer extension greater than 40 cm², 97% presented a new ulcer, 59% had ischemia, and 46% had severe infection (Table 1). The proportion of missing data for each variable is shown in Supplementary S3 Table.

### Differences between amputees and non-amputees

Patients with major amputation had a longer duration of illness and higher prevalence of comorbidities such as hypertension and prior stroke. Age, education level, and body mass index were similar between groups.

Regarding diabetic foot features, major amputation was associated with greater ulcer depth, severe infection, ischemia, and larger ulcer extension, but not with ulcer location or recurrence.

Laboratory variables showed an association between albumin < 2.5 g/dL and major amputation, whereas HbA1c, LDL cholesterol, and hemoglobin showed no significant association (Table 1).

When stratified by classification categories, most amputations were concentrated in the most advanced stages. For the Meggitt–Wagner system, 92% of amputations occurred in grades 4–5, where the amputation rates were 23.8% and 100%,

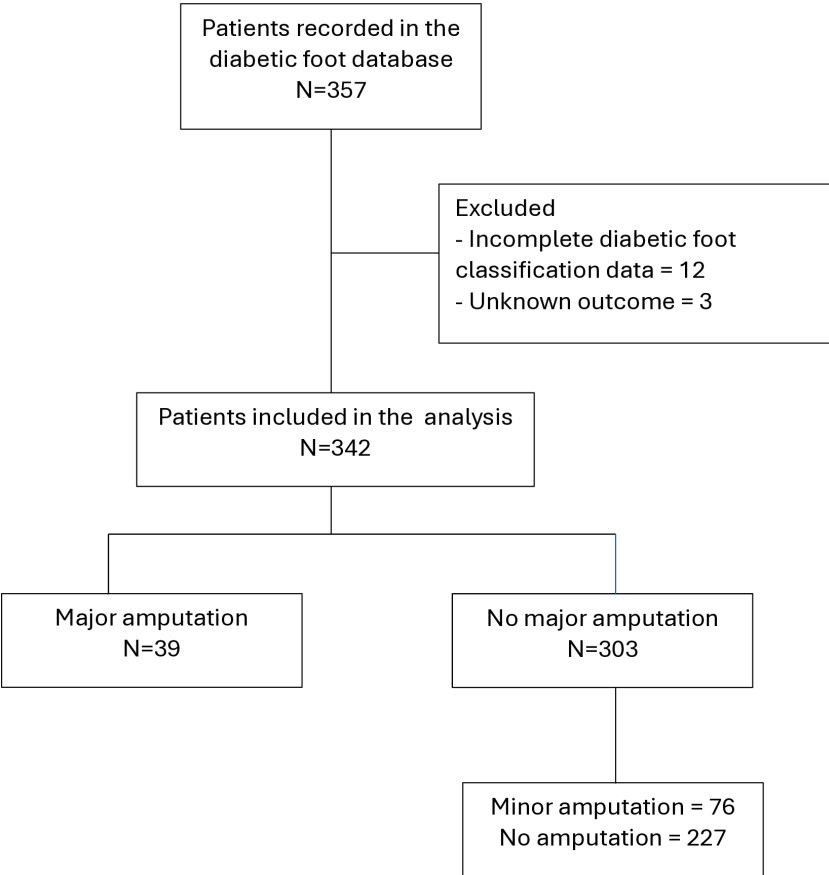

**Fig 1. Flow diagram of patient selection and inclusion in the validation cohort.**

respectively. In the Texas University classification, 87% of amputations were clustered in stage 3D, with an amputation rate of 30.9%. For SINBAD, 85% of amputations occurred in scores ≥5, with rates of 21.7% in score 5 and 20.0% in score 6. Finally, for the Saint Elian system, almost half of patients in the 21–25 category (47.6%) required major amputation, compared with only 2.7% in the 11–15 group and 16.5% in the 16–20 group (Supplementary S4 Table).

### Risk of major amputation

MW grades 4 or 5 increased the risk of major amputation 14.7 times (RR 15.7; 95% CI 4.9 to 50.2; p < 0.001) compared to the combined categories 1–3. TU stage 3D increased the risk 13.3 times (RR 14.3; 95% CI 5.7 to 35.7; p < 0.001) compared to the rest of the combined stages. When analyzing the three individual components of TU, only ischemia had sufficient cases to calculate risk (RR 4.59; 95% CI 1.84 to 11.48). A SINBAD score of 5 increased the risk 6.88 times (RR 7.88; 95% CI 3.11 to 19.9) compared to scores less than 4. Finally, an SE score greater than 20 increased the risk 31.2 times (95% CI 10.0 to 103.6) compared to scores less than 16 (Table 2).

### Prognostic capacity

Seven ROC curves corresponding to four classification systems—SE, MW, TU (3D stage, depth, ischemia, infection), and SINBAD—were analyzed (Fig 2).

**Table 1. Incidence of major amputation by clinical and epidemiological characteristics of patients with diabetic foot.**

| | Total N = 342 | Major amputation N = 39 | No major amputation N = 303 | P Value |
|---|---|---|---|---|
| **Demographics** | | | | |
| **Age (years)**, n(%) | | | | |
| <60 years | 167 (48.8) | 14 (35.9) | 153(50.5) | 0.086 [a] |
| >60 years | 175 (51.2) | 25 (64.1) | 150 (49.5) | |
| **Sex, n(%)** | | | | |
| Female | 117 (34.0) | 9 (23.08) | 108(35.6) | 0.119 [a] |
| Male | 225 (66.0) | 30 (76.9) | 195 (64.4) | |
| **Degree of instruction n(%)** | | | | |
| Elementary or less | 95 (34.9) | 14 (41.2) | 81 (34.1) | 0.712 [a] |
| High school | 169 (62.1) | 19 (55.9) | 150 (63.0) | |
| Superior | 8 (3.0) | 1 (2.9) | 7 (2.9) | |
| **Clinical history** | | | | |
| **Body mass index**, n(%) | | | | |
| <25 | 110 (45.1) | 13 (56.5) | 97 (43.9) | 0.155 [d] |
| 25 -29.9 | 98 (40.2) | 5 (21.7) | 93 (42.1) | |
| >30, | 36 (14.7) | 5 (21.7) | 31 (14.03) | |
| **Time of diabetes (years)**, n(%) | | | | |
| <10 years | 116 (38.3) | 9 (23.7) | 107 (40.4) | 0.048 [b] |
| >=10 years | 187 (61.7) | 29 (76.3) | 158 (84.5) | |
| **Previous diabetes treatment** | | | | |
| Diet only | 30 (9.6) | 5 (15.2) | 25 (8.9) | 0.231 [a] |
| Oral antidiabetics only (OAD) | 155 (49.2) | 12 (36.4) | 143 (50.7) | |
| Insulin +/- OAD | 130 (41.3) | 16 (48.5) | 114(40.3) | |
| **Glomerular Filtration Rate <60ml/min, n(%)** | | | | |
| >=60 ml/min | 227 (80.2) | 26 (70.3) | 201 (81.7) | 0.104 [a] |
| <60 ml/min | 56 (19.8) | 11 (29.7) | 45 (18.3) | |
| **Hypertension, n(%)** | | | | |
| **No** | 212 (62.4) | 14 (35.9) | 198 (65.8) | <0.001 [a] |
| **Yes** | 128 (38.1) | 25 (64.1) | 103 (34.2) | |
| **Coronary disease, n(%)** | | | | |
| No | 319 (98.8) | 39 (100) | 280 (98.6) | 0.456 [b] |
| Yes | 4 (1.2) | 0 (0.0) | 4(1.4) | |
| **Stroke, n(%)** | | | | |
| No | 301 (96.2) | 33 (84.6) | 268 (97.8) | <0.001 [a] |
| Yes | 12 (3.9) | 6 (15.4) | 6 (2.2) | |
| **Previous major amputation, n(%)** | | | | |
| No | 291(93.0) | 30 (88.2) | 261 (93.5) | 0.253 [b] |
| Yes | 22 (7.0) | 4 (11.8) | 18 (6.5) | |
| **Foot characteristics** | | | | |
| **Type of ulcer, n(%)** | | | | |
| New | 324 (96.4) | 38 (97.4) | 286 (96.3) | 0.718 [b] |
| Relapse | 12 (3.6) | 1 (2.6) | 11 (3.7) | |
| **Extention, n(%)** | | | | |
| <10 cm | 155 (45.3) | 3 (7.7) | 152 (50.2) | <0.001 [d] |
| 10-39.9 cm | 97 (28.4) | 6 (15.4) | 91 (30.0) | |

*(Continued)*

**Table 1.** (Continued)

| | Total N = 342 | Major amputation N = 39 | No major amputation N = 303 | P Value |
|---|---|---|---|---|
| >40 cm | 90 (26.3) | 30 (76.9) | 60 (19.8) | |
| **Depth, n(%)** | | | | |
| Superficial | 125 (36.6) | 0 (0.0) | 125 (41.3) | <0.001 [d] |
| Tendon or ligament | 58 (16.9) | 0 (0.0) | 58 (19.1) | |
| Articular join – Bone | 159 (46.5) | 39 (100) | 120 (39.6) | |
| **Location, n(%)** | | | | |
| Forefoot | 254 (74.3) | 30 (76.9) | 224 (73.9) | 0.687 [a] |
| Midfoot or hindfoot | 88 (25.7) | 9 (23.1) | 79 (26.1) | |
| **Ischemia, n(%)** | | | | |
| No | 138 (40.4) | 5 (12.9) | 133 (43.9) | <0.001 [a] |
| Yes | 204 (59.7) | 34 (87.2) | 170 (56.1) | |
| **Infection (IDSA), n(%)** | | | | |
| None | 91 (26.7) | 0 (0.0) | 91 (30.0) | <0.001 [d] |
| Mild | 103 (30.1) | 1 (2.6) | 102 (33.7) | |
| Moderate | 123 (36.0) | 20 (51.3) | 103 (34.0) | |
| Severe | 25 (7.3) | 18 (46.2) | 7 (2.3) | |
| **Lab Values** | | | | |
| **HbA1c %, n(%)** | 9.6±2.8 | | | |
| <7% | 48 (20.1) | 5 (20.0) | 43 (20.1) | 0.991 [a] |
| ≥ 7 | 191 (79.9) | 20 (80.0) | 171 (79.9) | |
| **LDL (mg/dL), n(%)** | 107.7±37.4 | | | |
| <100 | 84 (42.4) | 10 (43.5) | 74 (42.3) | 0.913 [a] |
| ≥ 100 | 114 (57.6) | 13 (56.5) | 101 (57.7) | |
| **Hb (g/dL)** | 11.2±2.1 | | | |
| <8 | 258 (75) | 36 (92.3) | 222 (73.2) | 0.005 [b] |
| >= 8 | 84 (25) | 3(7.7) | 81 (26.8) | |
| **Albumin (g/dL), n(%)** | 3.5±2.2 | | | |
| < 2.5 | 15 (12.1) | 6 (28.6) | 9 (8.7) | 0.011[a] |
| ≥ 2.5 | 109 (87.9) | 15 (71.4) | 94 (91.3) | |
| **Hospitalizations** | | | | |
| **Hospitalization status, n(%)** | | | | |
| Inpatient | 109 (31.9) | 39 (100) | 70 (23.1) | <0.001 [b] |
| Outpatient | 233 (68.1) | 0 (0.0) | 233 (76.9) | |
| **Lenght of hospital stay (days)** | | | | |
| Median [IQR] | 17 [12–27] | 22 [14–32] | 14.5 [10–20] | 0.002 [d] |
| **Minor amputation** | 76 (22.2) | 0 | 76 (25.1) | |

Major amputation defined as any amputation above the ankle. Percentages calculated within each category. All patients had type 2 diabetes mellitus. Statistical tests: [a] Chi squared. [b] Fisher test. [c] T Student. [d] U of Matt Whitney. IQR: Interquartile range.

Areas under the ROC curve (AUC) with 95% confidence intervals: Saint Elian, 0.900 (0.858–0.942); University of Texas (UT) 3D stage, 0.811 (0.752–0.869); Meggitt–Wagner, 0.805 (0.770–0.839); UT depth, 0.802 (0.774–0.829); SINBAD, 0.747 (0.693–0.802); UT ischemia, 0.655 (0.595–0.715); UT infection, 0.650 (0.624–0.676). The diagonal line represents chance discrimination.

Table 2. Incidence and relative risk of major amputation according to categories of the four diabetic foot classifications systems.

| | Major amputation | No major amputation | Relative risk | CI 95% | P Value |
|---|---|---|---|---|---|
| **Meggitt -Wagner scale, n(%)** | | | | | |
| 1-3 | 3 (1.6) | 191 (98.5) | 1.0 | | |
| 4-5 | 36 (24.3) | 112 (75.7) | 15.7 | 4.91–50.2 | 0.001 |
| **Texas University -Depth, n(%)** | | | | | |
| Superficial | 0 (0.0) | 125 (100.0) | NC | | |
| Tendon or ligament | 0 (0.0) | 58 (100.0) | NC | | |
| Bone | 39 (24.5) | 120 (75.5) | | | |
| **Texas University – infection, n(%)** | | | | | |
| No | 0 (0.0) | 91 (100.0) | NC | | |
| Yes | 39 (15.5) | 212 (84.5) | | | |
| **Texas University – ischemia, n(%)** | | | | | |
| No | 5 (3.6) | 133 (96.4) | 1.00 | | |
| Yes | 34 (16.7) | 170 (83.3) | 4.59 | 1.84-11.48 | 0.001 |
| **Texas University – 3D, n(%)** | | | | | |
| 1a – 3c | 5 (2.2) | 227 (97.8) | 1.0 | | |
| 3d | 34 (30.9) | 76 (69.1) | 14.3 | 5.75–35.7 | <0.001 |
| **Sinbad score, n(%)** | | | | | |
| 1-4 | 5 (2.8) | 177 (97.3) | 1.00 | | |
| 5 | 26 (21.7) | 94 (78.3) | 7.88 | 3.11–19.9 | <0.001 |
| 6 | 8 (20.0) | 32 (80.0) | 7.28 | 2.50–21.1 | <0.001 |
| **Saint Elian score, n(%)** | | | | | |
| <16 | 3 (1.5) | 200 (98.5) | 1.0 | | |
| 16 −20 | 16 (16.5) | 81 (83.5) | 11.1 | 3.32–37.4 | <0.001 |
| >20 | 20 (47.6) | 22 (52.4) | 32.2 | 10.0–103.6 | <0.011 |

Values are expressed as number (percentage). NC = not calculable due to absence of events in one category; CI = confidence interval. Relative risks, 95% CIs, and p-values were obtained using Poisson regression with robust variance.

The SE scale demonstrated the best accuracy compared to the others, with an AUROC of 0.900. Followed by TU 3D stage (AUROC = 0.811) and MW (AUROC = 0.805), both showing similar performance. Lastly, SINBAD had the lowest discriminative ability, with an AUROC of 0.747. Regarding the individual components of UT, depth showed an accuracy similar to MW (AUROC = 0.802). Meanwhile, the ischemia and infection components exhibited lower performances, with AUROCs of 0.655 and 0.650, respectively, being the lowest among all. When stratifying by admission status, prognostic performance was consistently lower among inpatients compared with the overall cohort. For inpatients (n = 109), the Saint Elian classification decreased to moderate discrimination (AUROC 0.74, 95% CI 0.65–0.82), while Meggitt–Wagner and SINBAD dropped to poor or very poor discrimination (AUROC 0.60, 95% CI 0.50–0.69, and AUROC 0.58, 95% CI 0.48–0.67, respectively). The University of Texas system also showed diminished accuracy in this subgroup (AUROC 0.66 for the 3D stage, 95% CI 0.56–0.75). By contrast, in outpatients no major amputations were observed during follow-up, preventing calculation of discrimination metrics in this group (Table 3).

Pairwise comparisons confirmed these rank orders. The Saint Elian (SE) scale discriminated significantly better than Texas University stage 3D (ΔAUC = 0.09, 95% CI 0.03–0.14; p = 0.001), Meggitt–Wagner (ΔAUC = 0.10, 0.06–0.14; p < 0.001), and SINBAD (ΔAUC = 0.16, 0.10–0.20; p < 0.001). TU 3D outperformed SINBAD (ΔAUC = 0.07, 0.02–0.10; p = 0.003) and did not differ from Meggitt–Wagner (ΔAUC ≈ 0.01; p > 0.80). Meggitt–Wagner showed only a borderline advantage over SINBAD (ΔAUC = 0.06, 0.00–0.11; p = 0.053). For the UT components, 'depth' performed similarly to

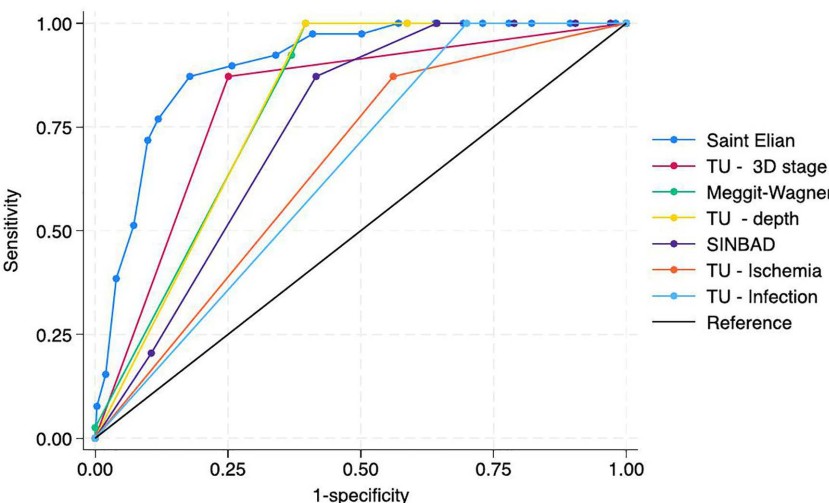

**Fig 2. Receiver operating characteristic (ROC) curves comparing the prognostic accuracy of diabetic foot classification systems for predicting six-month major amputation.**

Meggitt–Wagner (p ≈ 0.80) and showed a borderline advantage over SINBAD (p ≈ 0.05), whereas the 'ischemia' and 'infection' components performed significantly worse than SE, TU 3D, and Meggitt–Wagner (ΔAUC range −0.09 to −0.25; all p < 0.001) (Table 4).

In supplementary analyses, the statistical power of the AUROC comparisons was high. Specifically, the power exceeded 90% for the contrasts between Saint Elian versus Meggitt-Wagner and SINBAD, and reached 78% for Saint Elian versus Texas University (Supplementary S5 Table). This indicates that the study had sufficient precision to detect meaningful differences in discriminative performance between classification systems.

The performance of the classifications was informed thanks to the Youden index of each one. A Wagner score greater than or equal to 3 had an index of 52.7%, a Texas 3D stadium had an index of 62.1%, a SINBAD score greater than or equal to 5 had an index of 45.6%, and a Saint Elian score greater than or equal to 18 had an index of 69.4. In line with Youden's index (MW ≥ 3: 60.4%; TU 3D: 62.1%; SINBAD ≥ 5: 45.6%; SE ≥ 18: 69.4%), the best operating points also

**Table 3. Discriminative ability (AUROC with 95% CI) of diabetic foot classification systems for predicting major amputation, overall and stratified by hospitalization status.**

| Classification system | AUROC (95% CI) – All patients (n = 342) | AUROC (95% CI) – Inpatients (n = 109) |
|---|---|---|
| Saint Elian | 0.900 (0.864–0.930) | 0.741 (0.651–0.822) |
| Meggitt–Wagner | 0.805 (0.758–0.845) | 0.601 (0.498–0.689) |
| SINBAD | 0.747 (0.699–0.794) | 0.580 (0.480–0.672) |
| University of Texas – 3D | 0.811 (0.764–0.850) | 0.657 (0.564–0.749) |
| University of Texas – Depth | 0.802 (0.755–0.842) | 0.607 (0.507–0.698) |
| University of Texas – Ischemia | 0.655 (0.602–0.705) | 0.600 (0.498–0.689) |
| University of Texas – Infection | 0.650 (0.596–0.700) | 0.507 (0.407–0.602) |

AUROC: Area under the receiver operating characteristic curve; CI: Confidence interval. Estimates are presented for the full cohort (n = 342) and for the subgroup of hospitalized patients (n = 109). No amputations occurred among outpatients, so stratified AUROC estimates were not calculable in that group.

showed favorable sensitivity–specificity trade-offs. SE ≥ 18 achieved 87.2% sensitivity and 82.2% specificity (PPV 38.6%, NPV 98.0%; LR + 4.89, LR– 0.15). TU 3D showed 87.2% sensitivity and 74.9% specificity (PPV 30.9%, NPV 97.8%; LR + 3.47, LR– 0.17). MW ≥ 3 provided 100% sensitivity with 60.4% specificity (PPV 24.5%, NPV 100.0%; LR + 2.52, LR– 0.00), favoring rule-out. SINBAD ≥ 5 yielded 87.2% sensitivity and 58.4% specificity (PPV 21.3%, NPV 97.3%; LR + 2.09, LR– 0.21). Collectively, these thresholds indicate that SE ≥ 18 and TU 3D offer the strongest discrimination for identifying high-risk patients, while MW ≥ 3 maximizes sensitivity for screening (Table 5). In addition to these key thresholds, Supplementary S6 Table provides the complete evaluation of all possible cut-off points for each classification system.

As a complementary analysis, we also evaluated the performance of all seven classification systems for the broader outcome of "any amputation" (including both major and minor procedures). Meggitt–Wagner and the UT depth component showed the highest discrimination in the full cohort, whereas performance decreased substantially among hospitalized patients. In non-hospitalized patients, high-performing systems retained strong accuracy (AUC > 0.80). Full detailed results for this complementary sensitivity analysis are provided in Supplementary S7 Table.

## Discussion

### Main findings

In this external validation study, the Saint Elian classification showed excellent discriminative ability (AUROC 0.90), outperforming all other systems. Both the University of Texas stage 3D (AUROC 0.811) and the Meggitt–Wagner system (AUROC 0.805) achieved a good level of prognostic accuracy, while the SINBAD score (AUROC 0.747) demonstrated only moderate performance. Within the UT system, depth performed similarly to Wagner (AUROC 0.802, good), but the isolated ischemia (AUROC 0.655) and infection (AUROC 0.650) components had poor discriminatory power. These

**Table 4. Pairwise differences in areas under the ROC curve (AUC) for the prognostic accuracy of diabetic foot classifications in predicting six-month major amputation.**

|  | Saint Elian AUC 0.900 | Meggit-Wagner AUC 0.805 ¿ | SINBAD AUC 0.747 |
|---|---|---|---|
| **Texas University -3D Stage** AUC 0.811 | −0.09 (- 0.14 to −0.03); **p = 0.001** | 0.07 (- 0.06 to 0.05) p = 0.842 | 0.07 (0. 02 to 0.10); **p = 0.003** |
| **Texas University – Depth}** AUC 0.802 | −0.10 (−0.14 to −0.05); **p < 0.001** | 0.003 (−0.024 to 0.018); P = 0.803 | 0.06 (−0.11 to 0.006); 0.052 [a] |
| **Meggitt-Wagner** AUC 0.805 | −0.10 (−0.14 to − 0.06); **p < 0.001** | ------ | 0.06 (- 0.001 to 0.11); p = 0.053 |
| **SINBAD** AUC 0.747 | −0.16 (- 0.20 to − 0.10); **p < 0.001** | 0.06 (- 0.001 to 0.11); p = 0.053 | ------ |
| **Texas University - Ischemia** AUC 0.655 | −0.25 (−0.30 to −0.18); **p < 0.001** | −0.15 (−0.21 to −0.08); **p < 0.001** | − 0.09 (−0.14 to −0.05) **p < 0.001** |
| **Texas University – Infection** AUC 0.650 | −025 (−0.29 to −0.20; **p < 0.001** | − 0.15 (−0.19 to −0.12); **p < 0.001** [a] | −0.09 (−0.14 to −0.04); **p < 0.001** |

Values represent the absolute difference in AUC between scales with corresponding 95% confidence intervals and p-values (DeLong test for correlated ROC curves).

**Table 5. Prognostic performance of diabetic foot classification cut-off points for predicting six-month major amputation.**

| | Sensitivity (95% CI) | Specificity (95% CI) | PPV (95% CI) | NPV (95% CI) | LR [+] (95% CI) | LR [-] (95% CI) | Youden index |
|---|---|---|---|---|---|---|---|
| **Meggitt-Wagner** | | | | | | | |
| ≥ 3 | **100.0 (91-100)** | **60.4 (54.6-65.9)** | **24.5 (18.1-32)** | **100.0 (98-100)** | **2.52 (2.2-2.9)** | **0.00** | **60.4** |
| ≥ 4 | 92.3 (79.1-98.4) | 63.0 (57.3-68.5) | 24.3 (17.7-32.1) | 98.5 (95.5-99.7) | 2.49 (2.1-2.97) | 0.12 (0.041-0.363) | 55.3 |
| **TU Depth** | | | | | | | |
| ≥ 2 | 100 (91-100) | 41.3 (35.7-47) | 17.9 (13.1-23.7) | 100.0 (97.1-100) | 1.70 (1.55-1.87) | 0.00 | 41.3 |
| ≥ 3 | **100 (91-100)** | **60.4 (54.6-65.9)** | **24.5 (18.1-32)** | **100.0 (98-100)** | **2.52 (2.2-2.9)** | **0.00** | **60.4** |
| **TU Ischemia** | | | | | | | |
| Yes | 87.2 (72.6-95.7) | 43.9 (38.2-49.7) | 16.6 (11.8-22.5) | 96.4 (91.7-98.8) | 1.55 (1.33-1.82) | 0.29 (0.12-0.66) | 31.1 |
| **TU infection** | | | | | | | |
| Yes | 100.0 (91-100) | 30.0 (24.9-35.5) | 15.5 (11.3-20.6) | 100.0 (96-100) | 1.43 (1.33-1.54) | 0.00 | 30.0 |
| **TU 3D** | | | | | | | |
| 3D | **87.2 (72.6-95.7)** | **74.9 (69.6-79.7)** | **30.9 (22.4-40.4)** | **97.8 (95-99.3)** | **3.47 (2.76-4.37)** | **0.17 (0.07-0.38)** | **62.1** |
| ≥ 4 | 100.0 (91-100) | 35.6 (30.2-41.3) | 16.7 (12.1-22.1) | 100.0 (96.6-100) | 1.55 (1.43-1.69) | 0.00 | 35.6 |
| ≥ 5 | **87.2 (72.6-95.7)** | **58.4 (52.6-64)** | **21.3 (15.2-28.4)** | **97.3 (93.7-99.1)** | **2.09 (1.75-2.51)** | **0.21 (0.09-0.5)** | **45.6** |
| **Saint Elian** | | | | | | | |
| ≥ 10 | 100.0 (91-100 | 22.1 (17.6-27.2) | 15 (10.3-18.9) | 100.0 (95.6-100) | 1.37 (1.28-1.47) | 0.00 | 27.1 |
| ≥ 15 | 97.4 (86.5-99.9) | 59.1 (53.3-64.7) | 23.5 (17.2-30.7) | 99.4 (96.8-100) | 2.38 (2.06-2.75) | 0.04 (0.06-0.30) | 56.5 |
| ≥ 17 | 89.7 (75.8-97.1) | 74.3 (68.9-79.1) | 31.0 (22.6-40-4) | 98.3 (95.6-99.5) | 3.48 (2.8-4.3) | 0.13 (0.05-0.35) | 64.0 |
| ≥ 18 | **87.2 (72.6-95.7)** | **82.2 (77.4-86.4)** | **38.6 (28.4-49.6)** | **98.0 (95.5-99.4)** | **4.89 (3.73-6.41)** | **0.15 (0.68-0.35)** | **69.4** |
| ≥ 20 | 71.8 (55.1-85) | 90.1 (86.2-93.2) | 48.3 (35-61.8) | 96.1 (93.2-98.1) | 7.25 (4.9-10.7) | 0.31 (0.18-0.51) | 61.9 |

Values shown are sensitivity, specificity, positive predictive value (PPV), negative predictive value (NPV), positive likelihood ratio (LR+), negative likelihood ratio (LR–), and Youden index, each with corresponding 95% confidence intervals. Analyses were conducted in the overall cohort. Cut-off points were selected based on discriminative performance and clinical interpretability.

findings indicate that multidimensional systems such as SE and UT provide more robust prognostic information than single-dimension assessments, highlighting their potential value in clinical decision-making for diabetic foot care.

## Comparison with previous studies

Recent evidence highlights the variability and methodological limitations in the validation of diabetic foot ulcer classifications. A systematic review and meta-analysis conducted in Latin America and the Caribbean found that the Wagner system at grade 3 threshold achieved the highest sensitivity, while the Saint Elian classification showed the highest specificity, although the certainty of the evidence was rated very low due to heterogeneity and methodological weaknesses across studies [25]. Complementing these findings, a global systematic review identified 28 different systems applied in 149 studies, noting that most classifications—including Wagner, University of Texas, WIfI, PEDIS, and SINBAD—were validated in only a few studies each, with low or very low certainty of evidence, and no consensus on a universally applicable tool [26]. These results underscore the fragmented nature of the evidence base and the need for more rigorous, standardized validation efforts across diverse healthcare settings.

## Saint Elian classification

At the Latin American level, our results are consistent with an Argentine study that followed patients with new-onset diabetic foot ulcers for five months. This study reported an AUC of 0.893 for major amputation using the Saint Elian classification with a cut-off point greater than 18 [27]. Similarly, a Chinese study found that a Saint Elian score

greater than 17 reduced the probability of cure by 24%, although it did not report statistics for major amputation alone due to a small sample size [28]. The Saint Elian system, composed of ten variables, is theoretically better suited to predict major amputation; however, the inclusion of multiple variables increases interobserver variability and may lead to inconsistent results. This highlights the importance of training and standardized validation procedures.

## Texas University classification

Regarding the UT classification, a Philippine study reported an AUC of 0.785 for grade and 0.575 for stage [29]. In contrast, in our analysis, the University of Texas classification showed the second-best performance (AUC = 0.81) for major amputations. This difference may be explained by our dichotomous analysis, focused specifically on the presence of stage 3D, due to its high prevalence in our hospital population. The Texas system does not yield a single score but instead assigns patients into one of 16 categories based on four grades and four stages. In practice, the grade assesses lesion depth, while the stage identifies infection or ischemia [16]. An expanded version, the WIfI classification, was introduced nine years ago and also incorporates infection and ischemia categorization into a single outcome [30]. However, we were unable to evaluate this system due to limitations in vascular assessment.

## Wagner classification

The Wagner classification demonstrated intermediate prognostic performance. It primarily assesses ulcer depth and includes only a partial assessment of peripheral arterial disease. In cases with necrosis, it cannot distinguish whether the cause is infection or ischemia. As one of the earliest and most widely used classification systems, Wagner's utility in research is limited by its lack of detail regarding infection severity or ischemic status [31]. Nonetheless, in our study population, which included many patients with deep lesions and necrosis, this system had the third-best performance.

## SINBAD classification

The SINBAD classification, composed of six dichotomized variables, had the lowest performance (AUC = 0.74). While this system is endorsed by the International Working Group on the Diabetic Foot (IWGDF) for communication among healthcare providers and outcome comparisons across institutions. Its binary structure and fixed cut-offs may limit its discriminative power in high-risk populations such as hospitalized patients [32].

## Theoretical and practical implications

The Wagner system laid the foundation for prognostic assessment in diabetic foot, and subsequent tools—University of Texas, SINBAD, and Saint Elian—incorporated additional dimensions such as depth, infection, and ischemia to improve accuracy. In our setting, Saint Elian likely performed better because its multidimensional structure captures anatomical distribution, aggravating factors (ischemia, infection, edema), and contributing factors (depth, ulcer area, healing phase), providing a more comprehensive representation of disease severity. Despite its simplicity, Wagner remains clinically useful because ulcer depth strongly correlates with necrosis and infection—key drivers of limb loss. [33].

Binary assessments of ischemia and infection, as used in components of the University of Texas classification, performed poorly because they oversimplify conditions that exist on a spectrum and fail to capture intermediate stages that influence prognosis in this sample [16]. The very high relative risks observed in advanced categories are explained by the disproportionate clustering of amputations among these patients, which magnifies effect estimates [34]. The lower discrimination seen in hospitalized patients compared with the overall cohort is consistent with spectrum bias: inpatients typically present with more severe and homogeneous disease, reducing the ability of scoring systems to separate high- from low-risk cases [35].

At the hospital level, adopting Saint Elian or the University of Texas system as primary screening tools could help clinicians promptly identify patients at highest risk for major amputation, enabling intensified surveillance, targeted vascular evaluation, and timely multidisciplinary interventions. Wagner at grade ≥3 may function as a high-sensitivity rule-out threshold at first contact, while SINBAD at score ≥5 can facilitate communication and auditing across services. Other scores such as WIfI (Wound, Ischemia, and foot Infection) could not be evaluated in this cohort because they require systolic pressure measurements (ankle–brachial index, toe–brachial index, or transcutaneous oxygen), which were not systematically available; this highlights the importance of aligning tool selection with feasible diagnostic practices [36]. Finally, training and standardized application are essential to limit interobserver variability and ensure reproducible, clinically meaningful use of any classification system.

## Public health importance

The classifications can have different uses in research, auditing, clinical monitoring and prognosis. The estimates found in this study may change depending on the population evaluated, the resources of the health system and the existence of a diabetic foot management program.

At the primary level, the Peruvian diabetic foot guidelines consider only the Wagner classification for reference purposes [37]. The International Working Group on the Diabetic Foot (IWGDF 2023) guidelines recommend the use of SINBAD because it is simple and easy to use, and could be an alternative to improve the communication and statistics [38].

Our data suggest that in referral hospitals with a heavy burden of advanced disease, Saint Elian and University of Texas systems,may offer superior prognostic value, complementing—rather than replacing—tools used in primary care.

## Strengths and limitations

This study has several limitations that should be acknowledged. First, ischemia was assessed using qualitative Doppler waveform analysis rather than toe–brachial or ankle–brachial indices, which were often not feasible due to extensive lesions, edema, or pain. Although this approach is supported in diabetic foot care when pressure measurements are unreliable, it may introduce operator-dependent variability. Second, because outcome ascertainment relied primarily on surgical records from the index hospital, amputations performed at other institutions may not have been captured. This incomplete external capture would most likely lead to an underestimation of the true incidence of major amputation (bias toward underascertainment). However, the likelihood of missed amputations is expected to be low in this setting. María Auxiliadora Hospital is the only high-complexity public referral center for diabetic foot care in the catchment area, serving predominantly low-income populations covered by the national subsidized insurance system. Given the limited accessibility to private surgical services and the absence of nearby public facilities with comparable capability for major limb salvage or amputation, it is unlikely that a substantial number of patients would have undergone amputations elsewhere. Therefore, although the direction of this bias would lean toward underestimation, its magnitude is likely small. Third, the study was conducted in a single high-volume referral center, which may limit generalizability to primary care settings or hospitals with different patient profiles, resource constraints, or referral patterns. Fourth, although the dataset contained complete information for all variables required to compute the four classification systems and the primary outcome, several clinical and laboratory covariates had varying degrees of missingness, which restricted the ability to explore additional adjusted or multivariable analyses. Finally, interobserver variability in applying multidimensional classification systems—particularly Saint Elian—was not formally measured and may have introduced non-differential misclassification, potentially biasing estimates toward the null..

Strengths include a consecutive real-world cohort with six-month follow-up, application of four widely used systems, prespecified handling of sparse categories to stabilize estimates, and comprehensive operating characteristics (sensitivity, specificity, predictive values, likelihood ratios, and Youden indices) across the full range of scores and cut-offs (see S3 Table).

## Conclusion

In this external-validation study from a Peruvian referral hospital, the Saint Elian and University of Texas classifications demonstrated good prognostic accuracy for predicting major amputation at six months, whereas the Meggitt–Wagner and SINBAD systems showed moderate performance. For practical use, patients with a Saint Elian score of 18 or higher or a University of Texas stage 3D should be considered at high risk and referred to specialized multidisciplinary care. Similarly, a Wagner grade of 3 or more or a SINBAD score of 5 or greater should prompt close monitoring and intensive management. Incorporating these thresholds into local care pathways, while adapting the choice of classification system to available resources and patient profiles, may strengthen risk stratification and improve limb-salvage outcomes in comparable health systems.

## Supporting information

**S1 Table. STROBE checklist for cohort studies.**
(DOCX)

**S2 Table. TRIPOD checklist for external validation studies.**
(DOCX)

**S3 Table. Missing data by variable.**
(DOCX)

**S4 Table. Incidence of major amputation across detailed categories of diabetic foot classification systems.**
(DOCX)

**S5 Table. Statistical power of pairwise comparisons between ROC curves of diabetic foot classifications.**
(DOCX)

**S6 Table. Prognostic discriminative capacity across the full range of categories and cut-off points of diabetic foot classifications for major amputation.**
(DOCX)

**S7 Table. AUROC (95% CI) of each classification system for predicting any amputation, overall and stratified by hospitalization status.**
(DOCX)

**S1 Dataset. Anonymized dataset used for the external validation of the Meggitt–Wagner, University of Texas, SINBAD, and Saint Elian classifications to predict major amputation in patients with diabetes treated at a public hospital in Peru.**
(XLSX)

## Author contributions

**Conceptualization:** Luis Alberto Gallardo-Alburqueque, Marlon Yovera-Aldana.

**Data curation:** Marlon Yovera-Aldana.

**Formal analysis:** Luis Alberto Gallardo-Alburqueque, Yudith Quispe-Landeo, Marlon Yovera-Aldana.

**Investigation:** Luis Alberto Gallardo-Alburqueque, Yudith Quispe-Landeo, Ann Chanamé-Marín, Marlon Yovera-Aldana.

**Methodology:** Luis Alberto Gallardo-Alburqueque, Yudith Quispe-Landeo, Ann Chanamé-Marín, Leonardo J. Uribe-Cavero, Marlon Yovera-Aldana.

**Supervision:** Ann Chanamé-Marín, Leonardo J. Uribe-Cavero, Marlon Yovera-Aldana.

**Validation:** Luis Alberto Gallardo-Alburqueque, Yudith Quispe-Landeo, Ann Chanamé-Marín, Leonardo J. Uribe-Cavero, Marlon Yovera-Aldana.

**Writing – original draft:** Luis Alberto Gallardo-Alburqueque, Marlon Yovera-Aldana.

**Writing – review & editing:** Luis Alberto Gallardo-Alburqueque, Yudith Quispe-Landeo, Ann Chanamé-Marín, Leonardo J. Uribe-Cavero, Marlon Yovera-Aldana.

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
