## [Decision Letter · Decision Letter 0]

13 Aug 2025

Dear Dr. Yovera-Aldana,

Thank you for submitting your manuscript to PLOS ONE. After careful consideration, we feel that it has merit but does not fully meet PLOS ONE’s publication criteria as it currently stands. Therefore, we invite you to submit a revised version of the manuscript that addresses the points raised during the review process.

We look forward to receiving your revised manuscript.

Kind regards,

Esha Arora, Ph.D.

Academic Editor

PLOS ONE

Journal Requirements:

Reviewers' comments:

Reviewer's Responses to Questions

**Comments to the Author**

1. Is the manuscript technically sound, and do the data support the conclusions?

Reviewer #1: Yes

Reviewer #2: Partly

2. Has the statistical analysis been performed appropriately and rigorously?

Reviewer #1: Yes

Reviewer #2: Yes

3. Have the authors made all data underlying the findings in their manuscript fully available?

Reviewer #1: Yes

Reviewer #2: No

4. Is the manuscript presented in an intelligible fashion and written in standard English?

Reviewer #1: Yes

Reviewer #2: Yes

Reviewer #1: This external validity study compares four international classification systems (Meggitt-Wagner, University of Texas, SINBAD, and Saint Elian) for predicting major amputations in patients with diabetic foot ulcers at public hospitals in Peru.

As an empirical study on diabetic foot management that lacks evidence in Latin America, this study has significant implications for public health and clinical practice.

However, to enhance the quality of the paper, the authors are requested to make several revisions and additions.

Supplemental tables 1, 2, and 3 also show important classifications and indicators. These data should also be mentioned in the main text.

Clearly state the reasons and methods for integrating specific grades in the Meggitt-Wagner and Texas University classifications.

In order to provide practical suggestions for applying the research results to clinical practice, present specific examples of the classification scores applied in clinical settings in the conclusion section.

Reviewer #2: 1. Is the manuscript technically sound, and do the data support the conclusions?

I have answered “partly” mostly because I believe the manuscript would benefit from some important clarifications pertaining to the methodology. Please refer to point 5 for specific elements.

2. Has the statistical analysis been performed appropriately and rigorously?

The statistical tests used appear both appropriate and rigorous for the work that has been undertaken.

3. Have the authors made all data underlying the findings in their manuscript fully available?

In the manuscript and supplemental documents at hand, it does not appear that the dataset is “fully available”. Although the authors mention that “All relevant data are within the manuscript and its Supporting Information files.”, I am not sure what is available within these documents is granular enough to qualify as “fully available”. I will leave it up to the editor to determine whether what is provided is sufficient enough to meet PLOS one’s policy.

4. Is the manuscript presented in an intelligible fashion and written in standard English?

As a general rule, the manuscript is well written, with only seldom areas that require possible additional work, from a language standpoint.

What could require more work is the structure of the manuscript. The authors present all the information that is necessary in a manuscript, but it seems often spread out erratically and makes it hard for the reader to follow. Some comments supporting this are provided in point 5.

5. Review Comments to the Author

Overall, I applaud the authors for attempting to validate wound classification in their Lima Hospital. I think there is value to their work in reinforcing the external validity of such models in a broad range of contexts. With this considered, I believe the manuscript would benefit from significant restructuring and clarification. Here are my comments following the received outline of the manuscript:

Abstract:

Line 51, materials and methods: I would encourage the authors to provide a summary of the patient cohort as detailed in the body of the manuscript, specifically as it pertains to inclusion criteria and timeline.

Line 57: “two thirds were male”: it may be best to put the absolute number of individuals and the corresponding proportion.

Lines 65-67: “Strengthening the training of multidisciplinary teams in referral centers is essential to ensure the effective application of these classification systems in clinical decision-making.” It is difficult to see how this sentence naturally follows the first sentence of the conclusion that comments on the quality of the performance of the diabetic foot classification systems.

Introduction:

Line 89: “In the local context” seems too broad. Without looking at the reference, it is difficult for the reader to determine whether the authors mean Peru as a whole, metropolitan Lima, or their hospital catchment area within metropolitan Lima.

Line 96: “Therefore” implies a connection between the previous and current sentences, but such connection is difficult to make. Perhaps removing the word altogether would work best.

Methods:

As a whole, I think the methods require significant restructuring and additional clarifications.

For the restructuring: (1) the general order is mostly appropriate, with only a few exceptions, but that need to be corrected; (2) the content within the sections does not always fit with the stated title. There are also some redundancies.

For the additional clarifications, the most important ones in my opinion pertain to (1) data extraction: and (2) measurement: it is not clear which healthcare provider was responsible for registering patient data that are essential for calculation of the scores; outcome measurement: was ipsilaterality relative to the ulcer at presentation accounted for? Was the outcome possible to capture outside of their healthcare institution?

Full details:

Line 105: “Study design and clinical scenario”: the authors initiate a description of their healthcare institution regarding referral pathways and cases, then proceed further describe treatment protocol diabetic foot management in a completely another section (“Diagnostic and therapeutic algorithm at the Maria Auxiladora hospital” ) more than 80 lines later. For the benefit of the reader, it could be best to ensure all information about the hospital from which the data is sourced is mentioned at the same place in the body of the text.

Line 113: “Population, sample and sampling”:

Where were suitable patients identified from? We come to understand later (e.g., “Preparation” section), that their institution has a diabetic foot database. Perhaps shifting the information from there would be appropriate.

What were the patient’s reason for seeking medical care? It may simply be their foot ulceration, as lines 188-189 seem to indicate, but this should be explicitly clear.

Line 114 “The study include all patients with diabetic foot […]”: how was the diagnosis of diabetic foot made? A lower extremity ulceration and a concomitant diagnosis of diabetes? Exclusion criteria on lines 119-120 seem to suggest that greater refinement of possible, but irrelevant etiologies was performed. How were ulceration etiologies teased out?

Lines 116-118: “[…] patients also needed to have complete clinical information […] to allow for scoring […]”: what healthcare provider was tasked with extracting such clinical information? They mention the “Diabetic Foot Unit” on line 190, but whether physicians, nurses, or other healthcare professionals within that unit were responsible for this needs to clearer.

Line 116: “[…] from the date of admission.”: this suggests inclusion of only inpatients, which is not the case in this study, as both in and outpatient health services were considered. using a better term to describe the index event is warranted. Perhaps "from the date of initiation of the index healthcare event" is more appropriate?

Line 121: “All patients who met the eligibility criteria were included in the study”: this may not need to be mentioned, as it may go without saying.

Line 126: “Diabetic foot classifications”: this section would benefit from a more detailed rationale for selecting those specific classification schemes. Some of this rationale is detailed later, but is likely best placed in this section.

Line 160: “Major amputation”: (1) Was ipsilaterality relative to the ulcer at presentation a consideration? Although hard to know the exact proportion, there may be some individuals whose limb loss at 6 months was contralateral to the ulceration they initially presented with. This could significantly confound the results; (2) Where was this captured? At the index hospital only, or was it possible to identify amputation happening in another institution? This is crucial information, and if the authors could not collect data beyond their institution, this could represent a significant limitation to the study.

Line 163: “Other variables”: was diabetes type a consideration?

Line 176: “Ischemia”: the use of toe-brachial index or, to a lesser extent because of the confounding effect of medial calcinosis, ankle-brachial index is arguably better than qualitative assessment of arterial waveform for an accurate description of an individual’s degree of ischemia. This limitation was acknowledged in the paper discussion.

Line 181: “Procedures”: one could argue the information in this section is best suited for an earlier mention in the methods section (perhaps within the “Study design and clinical scenario section”).

Lines 185-186: “Subsequently, the records were filtered and depurated according to predefined eligibility criteria for this study”: this statement appears somewhat intuitive. It may not be needed.

Line 186: “depurated”: this term is not appropriate to describe data collection.

Lines 201-203: the definition of ischemia is redundant at this point of the manuscript.

Line 207: “Statistic analysis”: *Statistical

Lines 211-212: it is atypical to refer to the results of the study in the methods section of a manuscript. I would suggest they indicate that they have performed power calculation, then in the results section refer the reader to the corresponding supplemental table.

Results

Although perhaps not possible given the smaller cohort size, stratifying analyses based on admission status would probably be important. Requiring admission most likely indicate more severe disease and increase the risk of limb loss considerably.

Line 238: general characteristics: a statement about the total number of patients would help the reader get a better idea of size of the patient cohort.

Line 293: Table 3 has a typo in the first row, third column (“¿”)

Discussion

The discussion needs re-structuring. Although most of the essential information is present, it seems to be spread out in different sections, sometimes seemingly inappropriately. Furthermore, some of the cited sources ([30]) do not appear in the bibliography. I would encourage the authors to double check the references to ensure they are accurate.

Full details: (note that line numbers appear to be missing from now on)

“Main findings”: terms like “good” and “moderate” are used to describe the model discriminative ability: what cutoffs were used to make such statements should be made explicit in the methods section.

“Theoretical and practical implications”: (1) I would encourage the authors how this could change practice within their hospital. Perhaps adopting one of the better performing models to screen for high risk individuals that require intensified surveillance is in the cards? Do their findings have more broad implications. (2) The section’s last paragraph would be best placed in their section about previous studies.

“Public health importance”: (1) line 5: “World guidelines” is very broad. I would encourage the author to explicitly state which guidelines they're referring to. Furthermore, the source is a cohort study, rather than any guidelines. The authors should cite the guidelines directly. (2) last line: source 30 is not mentioned in the manuscript references.

“Strength and limitations”: lines 2-3 from the bottom: “doctors of the endocrinology service carried out an active search for the outcomes”: whether they were able to accurately extract outcome occurrence in a healthcare institution other than theirs will be very important information and, as detailed above, will need to be described in the methods of the paper.

**Do you want your identity to be public for this peer review?** For information about this choice, including consent withdrawal, please see our Privacy Policy

Reviewer #1: No

Reviewer #2: No

---

## [Author Response · Author response to Decision Letter 1]

19 Sep 2025

Response to Reviewers

Manuscript ID: PONE-D-25-33165

Title: External validation of the Meggitt-Wagner, Texas University, SINBAD, and Saint Elian classifications for predicting major amputation in patients with diabetes at a public hospital in Peru

Dear Editor and Reviewers,

We are grateful for the constructive feedback and careful evaluation of our manuscript. We have revised the paper thoroughly to address all the comments provided by the reviewers and the academic editor. In the revised version, we have restructured specific sections of the discussion, clarified methodological details, improved the presentation of tables and figures, and expanded our interpretation of the findings in both clinical and public health contexts.

Below, we provide a detailed point-by-point response to each observation, indicating the changes made in the manuscript and justifying them where appropriate. All modifications have been incorporated into the revised version, which is submitted both with tracked changes and as a clean copy, as per journal requirements.

We thank the reviewers and the editorial team for their thoughtful comments, which have substantially improved the clarity and robustness of our work.

Sincerely,

Marlon A. Yovera-Aldana, MD, MSc

Reviewer #1:

1. Supplemental tables 1, 2, and 3 also show important classifications and indicators. These data should also be mentioned in the main text.

We appreciate the reviewer’s suggestion. In the original version, the main text reported only overall incidence, relative risks, and AUROC values, while the detailed statistical power, incidence by specific categories, and sensitivity/specificity parameters were presented exclusively in the supplementary tables. In the revised manuscript, we have incorporated the key findings from these tables into the Results section. Specifically, from S1 Table we added that the statistical power of the comparisons between curves exceeded 90% for SE versus MW and SINBAD, and was 78% for SE versus TU 3D. From S2 Table we included that most amputations occurred in MW grades 4–5 (92%), TU stage 3D (87%), SINBAD scores ≥5 (85%), and SE >20 (48%).

2. Clearly state the reasons and methods for integrating specific grades in the Meggitt-Wagner and Texas University classifications.

We thank the reviewer for this important observation. In the original version, the manuscript reported the results of the Meggitt-Wagner and Texas University classifications using integrated categories, but the rationale for grouping was not explicitly stated. In the revised version, we have clarified both the reasons and the methods used. Specifically, categories with very few or no cases of major amputation were collapsed with adjacent groups to ensure adequate statistical stability and valid estimations of risk. For the Meggitt-Wagner classification, grades 1–3 were combined because no amputations occurred in grades 1–2, and only three amputations were observed in grade 3. For the Texas University classification, most amputations were concentrated in stage 3D, while other stages had zero or very few events; therefore, categories were collapsed into “3D” versus “other stages” to allow meaningful risk estimates. These procedures were pre-specified to address sparse data issues and are commonly recommended in prognostic validation studies.

3. In order to provide practical suggestions for applying the research results to clinical practice, present specific examples of the classification scores applied in clinical settings in the conclusion section.

We appreciate this valuable suggestion. In the original version, the Conclusion highlighted the overall performance of the four classification systems but did not provide concrete examples for clinical use. We have revised the Conclusion to include specific thresholds that clinicians can apply at the bedside or in outpatient settings. For instance, we now indicate that patients with a Saint Elian score ≥18 or a Texas University stage 3D should be considered at high risk for major amputation and referred urgently to specialized multidisciplinary care. Similarly, patients with a Wagner grade ≥3 or a SINBAD score ≥5 should be closely monitored and prioritized for intensive management. These additions provide clear and practical guidance for clinical decision-making in resource-limited settings.

Reviewer #2:

Abstract:

4. Line 51, materials and methods: I would encourage the authors to provide a summary of the patient cohort as detailed in the body of the manuscript, specifically as it pertains to inclusion criteria and timeline.

Original version (Abstract – Methods section):

“A retrospective cohort study was conducted, including patients with a follow-up period of up to six months from hospital admission. The primary outcome was the occurrence of major amputation. For each classification system, the area under the receiver operating characteristic curve (AUROC) was calculated.”

Revised version (Abstract – Methods section):

“We conducted a retrospective cohort study at María Auxiliadora Hospital, one of the few referral centers in Lima with a specialized Diabetic Foot Unit. The study period was January 2015 to December 2019. Eligible patients had a lower-limb ulcer, complete clinical data recorded within 48 hours of the index healthcare encounter (hospital admission or outpatient evaluation), and at least six months of follow-up. Patients with venous ulcers or pressure ulcers related to immobilization were excluded.. The primary outcome was major amputation, defined as any procedure above the ankle. Prognostic performance of the four systems was assessed using the area under the receiver operating characteristic curve (AUROC), sensitivity, specificity, predictive values, and the Youden index

5. Line 57: “two thirds were male”: it may be best to put the absolute number of individuals and the corresponding proportion.

Original version (Results – General characteristics):

“The median age was 62 years (IQR: 54–69), and two thirds were male.”

Revised version:

“The median age was 62 years (IQR: 54–69), and 229 of 342 patients (67.0%) were male.”

6. Lines 65-67: “Strengthening the training of multidisciplinary teams in referral centers is essential to ensure the effective application of these classification systems in clinical decision-making.” It is difficult to see how this sentence naturally follows the first sentence of the conclusion that comments on the quality of the performance of the diabetic foot classification systems.

We have restructured the section to provide a logical flow from study findings to clinical and public health implications.

Original version (Conclusion):

“The Saint Elian and University of Texas classifications demonstrated good prognostic accuracy for major amputation at six months, while the Meggitt-Wagner and SINBAD classifications showed moderate performance. Strengthening the training of multidisciplinary teams in referral centers is essential to ensure the effective application of these classification systems in clinical decision-making.”

Revised version (Conclusion):

“The Saint Elian and University of Texas systems showed the best prognostic accuracy, while Meggitt-Wagner and SINBAD performed moderately. These findings provide clinicians with clear cut-off points to identify high-risk patients and support early interventions in referral settings. Their effective use depends on multidisciplinary teams trained in diabetic foot management, who can apply the classifications consistently and integrate them into decision-making. At the hospital and public health level, selecting the classification system best suited to available resources and patient profiles may optimize care pathways and contribute to reducing amputation rates..

Introduction:

7. Line 89: “In the local context” seems too broad. Without looking at the reference, it is difficult for the reader to determine whether the authors mean Peru as a whole, metropolitan Lima, or their hospital catchment area within metropolitan Lima.

We thank the reviewer for highlighting the ambiguity of the phrase “in the local context.” We agree that the scope needed to be specified in order to improve clarity for international readers.

Original version(Introduction):

“In the local context, diabetic foot affects approximately 30% of patients attending healthcare facilities [10]. However, only 11.5% to 15% of these patients achieve adequate metabolic control, defined as optimal levels of blood glucose, arterial pressure, and lipid profile [11,12]. Furthermore, diabetic foot is the reason for hospital admission in one out of every five patients with diabetes in Peruvian hospitals [13].”

Revised version (Introduction):

In the Peruvian context, diabetic foot affects approximately 5.9% of patients attending healthcare facilities [9]. It is also accounts one out of every five hospital admissions among patients with diabetes in Peruvian hospitals [10]. Limitations in healthcare personnel training, infrastructure, and funding hinder the establishment of multidisciplinary teams dedicated to diabetic foot care.[11] Furthermore, only 11.5% to 15% of these patients achieve adequate metabolic control —defined as optimal levels of blood glucose, blood pressure, and lipid profile —which further contributes to suboptimal clinical outcomes [12,13].

8. Line 96: “Therefore” implies a connection between the previous and current sentences, but such connection is difficult to make. Perhaps removing the word altogether would work best.

We thank the reviewer for noting the awkward transition at the beginning of this sentence. We agree that “Therefore” implies a causal link that is not clearly established and that rewrite the last paragraph to improve clarity and readability.

Original version (Introduction):

“Therefore, validating prognostic tools in settings different from those in which they were originally developed is essential to ensure their accuracy and applicability. Local contextual factors may influence the predictive performance of clinical scoring systems for major amputation.”

Revised version (Introduction):

Most clinical scoring systems for diabetic foot were developed and validated in high-income countries, and their predictive performance may not translate directly to Latin American settings. Differences in patient profiles, healthcare pathways, and resource availability can influence prognostic accuracy, underscoring the need for external validation. To address this gap, we compared the discriminative capacity of four widely used classifications—Meggitt-Wagner (MW), University of Texas score (TU), SINBAD, and Saint Elian (SE)—to predict major amputation at six months among patients managed at the Diabetic Foot Unit of a national referral

Methods:

9. As a whole, I think the methods require significant restructuring and additional clarifications.

For the restructuring: (1) the general order is mostly appropriate, with only a few exceptions, but that need to be corrected; (2) the content within the sections does not always fit with the stated title. There are also some redundancies.

For the additional clarifications, the most important ones in my opinion pertain to (1) data extraction: and (2) measurement: it is not clear which healthcare provider was responsible for registering patient data that are essential for calculation of the scores; outcome measurement: was ipsilaterality relative to the ulcer at presentation accounted for? Was the outcome possible to capture outside of their healthcare institution?

We thank the reviewer for this thorough assessment. In the revised manuscript, we have restructured the Methods section to improve coherence, reduce redundancies, and ensure that content is aligned with the appropriate subheadings. Regarding data extraction, we now specify that physicians from the Diabetic Foot Unit were responsible for systematically registering clinical, laboratory, and ulcer-related information at admission. For outcome measurement, we clarified that only major amputations ipsilateral to the presenting ulcer were considered, and ascertainment was based on hospital surgical records complemented by active follow-up. To minimize loss of outcome data, patients were contacted by telephone when follow-up visits were missed. We also added contextual information: very few hospitals in Lima have a specialized Diabetic Foot Unit, so most patients with new lesions preferentially attend our center, where they are known and managed as continuing patients, which ensures expedited care. Although Peru did not have an electronic medical record system at the time of the study and patient files were maintained manually, the probability of missing outcomes due to patients moving or seeking care elsewhere was considered low, but this limitation has been explicitly acknowledged

Full details:

10. Line 105: “Study design and clinical scenario”: the authors initiate a description of their healthcare institution regarding referral pathways and cases, then proceed further describe treatment protocol diabetic foot management in a completely another section (“Diagnostic and therapeutic algorithm at the Maria Auxiladora hospital” ) more than 80 lines later. For the benefit of the reader, it could be best to ensure all information about the hospital from which the data is sourced is mentioned at the same place in the body of the text.

In the revised manuscript, we consolidated the description of the hospital setting and care pathways into a single subsection at the beginning of the Methods, titled “Study setting and clinical management.” This subsection now integrates both the description of referral pathways, patient flow, and the specialized Diabetic Foot Unit, along with the diagnostic and therapeutic algorithm followed at our hospital.

11. Line 113: “Population, sample and sampling”: Where were suitable patients identified from? We come to understand later (e.g., “Preparation” section), that their institution has a diabetic foot database. Perhaps shifting the information from there would be appropriate. What were the patient’s reason for seeking medical care? It may simply be their foot ulceration, as lines 188-189 seem to indicate, but this should be explicitly clear.

In the revised manuscript, we now specify that suitable patients were identified from the diabetic foot database maintained by the Endocrinology Service at María Auxiliadora Hospital, which prospectively records all patients managed by the Diabetic Foot Unit. We also clarified that the primary reason for seeking medical care in all included cases was the presence of a lower-limb ulcer, infection, or ischemia, with most patients attending the Unit at the first sign of a lesion because of the hospital’s specialized referral role. This information has been relocated to the “Population and data source” subsection to ensure consistency and improve clarity.

12. Line 114 “The study include all patients with diabetic foot […]”: how was the diagnosis of diabetic foot made? A lower extremity ulceration and a concomitant diagnosis of diabetes? Exclusion criteria on lines 119-120 seem to suggest that greater refinement of possible, but irrelevant etiologies was performed. How were ulceration etiologies teased out?

In the revised version, we clarified that the diagnosis of diabetic foot was defined as the presence of a lower-extremity ulcer in a patient with a confirmed diagnosis of diabetes mellitus. The diagnosis of diabetes was based on medical history and laboratory records. For ulcer etiologies, differentiation was performed clinically by the Diabetic Foot Unit at the time of admission: venous ulcers were excluded based on typical location, chronic venous insufficiency signs, and absence of neuropathic or ischemic features; pressure ulcers were excluded if located on the heel and associated with prolonged immobilization or bedridden status. Only patients with neuropathic, ischemic, or mixed diabetic ulcers were included. This refinement has been explicitly described in the “Population and eligibility criteria” subsection of the Methods

13. Lines 116-118: “[…] patients also needed to have complete clinical information […] to allow for scoring […]”: what healthca

---

## [Decision Letter · Decision Letter 1]

28 Oct 2025

Dear Dr. Yovera-Aldana,

Thank you for submitting your manuscript to PLOS ONE. After careful consideration and review by the experts, we suggest you to make minor revisions of the manuscript before we proceed. Please find attached the comments below for the reference.

We look forward to receiving your revised manuscript.

Kind regards,

Esha Arora, Ph.D.

Academic Editor

PLOS ONE

Journal Requirements:

Reviewers' comments:

Reviewer's Responses to Questions

**Comments to the Author**

Reviewer #1: All comments have been addressed

Reviewer #3: (No Response)

2. Is the manuscript technically sound, and do the data support the conclusions?

Reviewer #1: Yes

Reviewer #3: Yes

3. Has the statistical analysis been performed appropriately and rigorously?

Reviewer #1: Yes

Reviewer #3: I Don't Know

4. Have the authors made all data underlying the findings in their manuscript fully available?

Reviewer #1: Yes

Reviewer #3: Yes

5. Is the manuscript presented in an intelligible fashion and written in standard English?

Reviewer #1: Yes

Reviewer #3: Yes

Reviewer #1: (No Response)

Reviewer #3: This is a retrospective cohort study of patients with lower-limb ulcer and looking at major amputations as outcomes,

Please consider reporting using Strobe Guidelines as many important elements are missing.

Alternatively for clinical prediction, you can use Tripod Reporting guidelines

What % of data was missing and how was missing data being managed

What about minor amputations?

Why were they excluded?

**Do you want your identity to be public for this peer review?** For information about this choice, including consent withdrawal, please see our Privacy Policy

Reviewer #1: No

Reviewer #3: **Yes:**  Ang Yee Gary

---

## [Author Response · Author response to Decision Letter 2]

21 Nov 2025

Editor-in-Chief,

We thank the Editor and reviewers for the time dedicated to evaluating our manuscript and for the constructive feedback provided. We appreciate the positive comments and have revised the manuscript accordingly, incorporating additional analyses, clarifying methods, and improving the overall presentation.

A detailed point-by-point response is provided below, and all modifications in the manuscript are highlighted in red. We trust that the revised version satisfactorily addresses all comments.

Sincerely,

Marlon Yovera-Aldana, on behalf of all co-authors

Reviewer #3

Comment 1: " Please consider reporting using Strobe Guidelines as many important elements are missing. Alternatively for clinical prediction, you can use Tripod Reporting guidelines "

Response 1: We thank the reviewer for this observation. We have ensured full adherence to the STROBE guidelines for cohort studies and the TRIPOD statement for external validation studies. A dedicated “Reporting guidelines” subsection has been added in the Methods section, and the completed STROBE and TRIPOD checklists have been included as Supplementary Tables S1 and S2. In addition, several reporting elements required by these guidelines have been completed: (1) missing data proportions are now explicitly reported and summarized in a supplementary table; (2) the direction and potential magnitude of key limitations are now described; and (3) the study design section has been expanded to provide clearer methodological detail.

Comment 2: “What % of data was missing and how was missing data being managed?”

Response 2: We thank the reviewer for this comment. The proportion of missing data for each variable has been reported and summarized in Supplementary Table S3. Missingness ranged from 0% to 63.7% across variables. As detailed in the Methods section, no imputation procedures were applied; all analyses were performed using complete-case data for each variable. In addition, 15 patients were excluded because they had incomplete information regarding the outcome or one or more of the variables required to compute the diabetic foot classification scales (Figure 1).

Comment 3: “What about minor amputations? Why were they excluded?”.

Response 3: We thank the reviewer for this comment. The study was designed to evaluate the prognostic performance of the four diabetic-foot classification systems for major amputation, which was the predefined primary outcome. For this reason, minor amputations were not included in the primary analyses. The definition of the outcome has been clarified in the Methods section to specify that the “no major amputation” group includes both patients with no amputation and those who underwent minor amputation. The proportion of minor amputation has also been added to Figure 1 and Table 1.

Because different studies evaluate either major amputation or amputation at any level, we performed a complementary analysis using the broader outcome “any amputation” (major or minor combined). As this was not the primary objective of the study, the complete results of this complementary analysis are presented in Supplementary Table S7.

---

## [Decision Letter · Decision Letter 2]

14 Dec 2025

External validation of the Meggitt-Wagner, Texas University, SINBAD, and Saint Elian classifications for predicting major amputation in patients with diabetes at a public hospital in Peru.

PONE-D-25-33165R2

Dear Dr. Yovera-Aldana,

We’re pleased to inform you that your manuscript has been judged scientifically suitable for publication and will be formally accepted for publication once it meets all outstanding technical requirements.

Kind regards,

Xiaoen Wei

Academic Editor

PLOS One

Additional Editor Comments (optional):

Reviewers' comments:

Reviewer's Responses to Questions

**Comments to the Author**

Reviewer #1: All comments have been addressed

Reviewer #3: All comments have been addressed

2. Is the manuscript technically sound, and do the data support the conclusions?

Reviewer #1: Yes

Reviewer #3: Yes

3. Has the statistical analysis been performed appropriately and rigorously?

Reviewer #1: Yes

Reviewer #3: I Don't Know

4. Have the authors made all data underlying the findings in their manuscript fully available?

Reviewer #1: Yes

Reviewer #3: Yes

5. Is the manuscript presented in an intelligible fashion and written in standard English?

Reviewer #1: Yes

Reviewer #3: Yes

Reviewer #1: (No Response)

Reviewer #3: Thank you for making the suggested changes.

I have no further comments and I thank you for your effort in writing up this manuscript.

**Do you want your identity to be public for this peer review?** For information about this choice, including consent withdrawal, please see our Privacy Policy

Reviewer #1: No

Reviewer #3: **Yes:**  Ang Yee Gary

---

## [Editor Report · Acceptance letter]

PONE-D-25-33165R2

PLOS One

Dear Dr. Yovera-Aldana,

I'm pleased to inform you that your manuscript has been deemed suitable for publication in PLOS One. Congratulations! Your manuscript is now being handed over to our production team.

Kind regards,

on behalf of

Dr. Xiaoen Wei

Academic Editor

PLOS One